# Adolescent Awkwardness: Alterations in Temporal Control Characteristics of Posture with Maturation and the Relation to Movement Exploration

**DOI:** 10.3390/brainsci10040216

**Published:** 2020-04-05

**Authors:** Felix Wachholz, Federico Tiribello, Maurice Mohr, Steven van Andel, Peter Federolf

**Affiliations:** Department of Sport Science, University of Innsbruck, 6020 Innsbruck, Austriamaurice.mohr@uibk.ac.at (M.M.); steven.van-andel@uibk.ac.at (S.v.A.); peter.federolf@uibk.ac.at (P.F.)

**Keywords:** adolescent awkwardness, motor control, body height, automatization, principal component analysis, postural control, adolescents and adults, exploration, minimal intervention principle

## Abstract

A phenomenon called adolescent awkwardness is believed to alter motor control, but underlying mechanisms remain largely unclear. Since adolescents undergo neurological and anthropometrical changes during this developmental phase, we hypothesized that adolescents control their movements less tightly and use a different coordinative structure compared to adults. Moreover, we tested if emerging differences were driven by body height alterations between age groups. Using 39 reflective markers, postural movements during tandem stance with eyes open and eyes closed of 12 adolescents (height 168.1 ± 8.8 cm) and 14 adults were measured, in which 9 adults were smaller or equal than 180 cm (177.9 ± 3.0 cm) and 5 taller or equal than 190 cm (192.0 ± 2.5 cm). A principal component analysis (PCA) was used to extract the first nine principal movement components (*PM_k_*). The contribution of each *PM_k_* to the overall balancing movement was determined according to their relative variance share (*rVAR_k_*) and tightness of motor control was examined using the number of times that the acceleration of each *PM_k_* changed direction (*N_k_*). Results in *rVAR_k_* did not show significant differences in coordinative structure between adolescents and adults, but *N_k_* revealed that adolescents seem to control their movements less tightly in higher-order *PM_k_*, arguably due to slower processing times and missing automatization of postural control or potential increases in exploration. Body height was found to not cause motor control differences between age groups.

## 1. Introduction

During maturation, adolescents are suggested to experience a period of altered motor control called ‘*adolescent awkwardness*’ [1]. During this phase of development, it is assumed that adolescents experience neurological, multisensory and information processing changes resulting in less efficient postural control compared to adults [2]. While postural control deficits in 10–14-year-old individuals are clearly evident from reduced performance in standing balance and dynamic functional tests [3,4], underlying differences in the behavior of the control system remain largely unclear. Investigating movement strategies and differences in the temporal structure of how movement strategies are controlled during balance tasks, offers a novel approach for assessing differences in sensorimotor control between adolescents and adults. 

Postural control can be studied using an inverted pendulum model, characterized by its sway amplitude and sway frequency [5,6]. Efficient postural control develops during maturation from a ballistic behavior with large amplitude and low frequency sway motions seen in children, to a tighter control behavior with smoother and more frequent oscillations observed in adulthood [7]. This development, however, is not assumed to be linear [8]. During the phase of adolescent awkwardness, postural control abilities have been reported to stagnate or even decline [1]. Suggested mechanisms for this altered development in adolescents include (1) an underdeveloped ability to estimate an internal model of body orientation [4,9], potentially resulting from a growth spurt [10]; (2) slower movement detection times [11]; (3) mislaid processing mechanisms induced by extensive muscle activation [12]; and (4) the heavy reliance on visual sensory input due to a lack of automatized control strategies [13]. Considering the nature of these causes within the motor control system, it is perhaps unsurprising that a link has been shown between the age of peak height velocity (i.e., the year of the biggest growth in adolescence) and increased injury risk in sport [10,14,15]. Given this relevance, we aim to investigate some of the mechanisms of adolescent awkwardness, with a particular focus on postural movements. 

When analyzing postural movements, it is important to consider the role of variability in posture. Traditionally, variability in stance has been interpreted as error since any deviation from a perfect vertical would bring one closer to falling [16,17]. However, it has been argued that there could also be functional aspects to this variability [16], as variability might represent the exploration of the dynamics in the postural control system [18]. From the viewpoint of perceptual motor control of action, this exploration is a crucial component of ‘getting to know the limits of the system’, a process that is particularly important in periods when the system has undergone recent changes [19,20]. For instance, this effect has been shown in pregnant women who undergo quite rapid changes to body dimensions and need to experience these new dynamics in order to become proficient in their movements shown in decision making [21] or in gait patterns [22]. It could be reasoned that the time around peak height velocity in adolescence is a similar period of rapid change, increasing the need for exploration in the postural system. 

Considering such an increased need for exploration to occur in adolescents, it could manifest within their movement patterns. Following the ‘minimal intervention principle’ it could be expected that there is an emergent weighting of costs and benefits in the movement system [23,24]. That is, control of movement dimensions is subject to a ‘costs function’ that biases movement control to steer away from those dimensions that could potentially endanger the movement outcome or could add costs to the system (e.g., in terms of stability). From this, it could be reasoned that, for instance when people stand in tandem stance (one foot in front of the other), exploration would be expressed in the dimensions describing (more stable) anterior–posterior directions [25]. Further, exploration would be expressed to a lower extent in the dimensions that represent the (less stable) mediolateral directions, due to the greater stability cost during or after a developmental phase. 

When assessing effects of developmental stage and age on postural control, height needs to be considered as an effect-modifying variable. According to the inverted pendulum model, a taller pendulum will sway at a slower frequency and greater amplitude [5,6]. Hence, increased body height is suggested to impede balance and increase the fall risk [26,27,28]. Literature results are inconsistent, however, since there is conflicting evidence regarding the correlation between body height and postural control abilities [29,30]. Either way, when investigating effects of adolescence on postural control, it is important to consider effect modification by body height. 

Generally, measurements of balance are often limited to lower-dimensional analyses of body sway and/or center of pressure movement based on clinical scale or newer technologies like wireless devices [31]. It should be considered that the exploration function in balance could be difficult to distinguish in a variable that basically averages activity of the entire system. A tool that could analyze different components of the system could perhaps be more sensitive. Furthermore, such a tool could help to explain the above reported inconsistencies in the role of height in balance control. In this light, we reason that a potentially more useful method for examining postural control strategies is the analysis of 3D kinematic movement data through a principle component analysis (PCA) [32,33,34,35]. Recent studies successfully used PCA-determined variables to provide insight into principal movement strategies during standing balance tasks with a specific focus on the number of interventions that the postural control system deploys in each movement dimension to maintain balance. We counted as an intervention when the movement acceleration changed direction, since acceleration changes indicate changes in muscle activation. Variance in movement strategies between the use of dominant and non-dominant leg during one leg stance were found [36] and might be correlated with a higher injury risk in downhill skiers [37]. Alterations between adults and elderly were found in the number of interventions of the control system, whereas the elderly were suggested to control less tight in some, but not in every movement dimension during tandem stance [38].

The purpose of this study was to investigate differences in stable postural control between adolescents and adults during tandem stance through application of a PCA on 3D kinematic data quantifying full-body balancing movements. The primary goal was to compare (1) the coordinative structure of postural movements, specifically the contribution of different movement dimensions used by adolescents and adults during stance; and (2) the control of individual movement strategies. It was expected that adolescents show more variable movement strategies to maintain balance (hypothesis 1), specifically an increase in anterior–posterior movement is expected to account for exploration in a dimension with low task relevance following the minimal intervention principle. Furthermore, if differences exist between control strategies in adolescents and adults, we expect that adolescents show fewer postural interventions (less tight control) compared to adults as an effect of the increased exploration function (hypothesis 2). However, it should be noted that any significant differences found between adolescents and adults could potentially just be the effect of body height on coordination. An exploratory analysis was done to investigate this potentially confounding effect. 

## 2. Materials and Methods

### 2.1. Participants

A convenience data sample of 26 participants (Table 1), originally recorded for another project [39] was analyzed for the purpose of this study (Appendix A). Exclusion criteria were diagnosed injuries, concussion, or other neurological disorders within the last six months, as well as self-reported problems concerning joint, tendons or muscles. Further, in the current study only data sets were included in which participants were able to stand without any kind of visible compensatory movement that led to a change in position. The current study specifically aimed at comparing motor control between adults and adolescents in stable trials. Visible instabilities, which can be expected to trigger additional postural control mechanisms, were avoided. Therefore, making a small step, lifting the toe or taking away the hand off the hip led to exclusion. The reason for including only such trials was to ensure that only data of the tandem stance during “successful” motor control without the need of compensation were analyzed.

The included participants were divided into two groups for maturation status (12 adolescents and 14 adults). To test for the effect of body height, adults were further separated into two height groups; *n* = 9 smaller or equal than 180 cm, and *n* = 5 taller or equal than 190 cm (incidentally none of the recruited participants were between 181 cm and 189 cm, which was the main motivation behind this split). Volunteers were healthy and physically active persons. All three groups differed significantly (*p* < 0.05) in height and weight, only in age no differences occurred between the adult groups. In the Appendix A, a file with more detailed subject characteristics can be found.

### 2.2. Measurement Procedure

Participants were instructed to stand as still as possible on a marked area on the ground in tandem stance (one foot in front of the other, the toe of the back foot slightly touching the heel of the foot in front). Tandem stance was selected, because it was considered a moderately challenging balance exercise for both participant groups. Participants were free to decide which foot they want to place in front and were advised to choose the one on which they felt most confident while keeping balance. Two trials were conducted, always with the same foot in front. For the first trial, they were asked to focus their gaze on a target cross placed 5.5 m in front of them at 1.75 m height and to stand as still as possible for 60 s. Visible compensatory movements were avoided. Hence, making a step, lifting the toe or foot, or moving the hand off the hip led to the trial being excluded and the participant was asked to re-do the trial. For the second trial, they were asked to again stand as motionless as possible, but with closed eyes. The eyes-closed trials were conducted for only 30 s, since pilot testing had revealed problems (especially among adolescents) with maintaining balance for 60 s with eyes closed in tandem stance. Between the trials, the volunteers were given rest periods of at least 90 s, in which they were free to move around in the room. The order of the two trials was not randomized, since the eyes open trial was used to evaluate if participants are stable enough for the eyes closed trial as well. Therefore, we dispensed randomization to prevent falls in the eyes closed trial. 

### 2.3. Instrumentation

Kinematic movement data of the volunteers were recorded at 250 Hz. An 8-camera Vicon motion tracking system (Vicon Motion Systems Ltd., Oxford, UK) collected the 3D coordinates of 39 retro-reflective markers that were attached on the participant’s skin or tight sport-clothing using double-sided tape. The markers were positioned on anatomical landmarks in accordance with the “Full-Body Plug-In Gait” provided by Vicon. Modified sweatbands were used to attach the markers on the head and on the wrist. The Vicon software (Vicon Nexus, Version 2.2.3; Vicon Motion Systems Ltd., Oxford, UK) was used for reconstructing the marker trajectories. 

### 2.4. Data Processing

MatLab R2019b (The Mathworks Inc., Natick, MA, USA) and the “PManalyzer” [40] were used to process the kinematic data. In all trials, the first 5 s were omitted to avoid settle-in effects, nine asymmetrical markers were removed, and gaps in marker-trajectories were reconstructed using a PCA-based procedure [41,42]. Then, animated stick figures were created for each trial in original motion. These video-representations were again screened for movements listed in the exclusion criteria, e.g., making a small movement with one foot or slightly shifting a hand off the hips. If minor, yet visible, compensatory movements were recognized, data were excluded in this step to analyze ‘successful’ motor control only. For each trial, a 16 s period was extracted, in which participants were stable without any disallowed movements. 

Then, data of participants with the left foot in front (*n* = 6) were mirrored and relabeled, such that for the analysis all participants appeared to stand with their right foot in front [38]. The normalization procedure before performing the PCA contained three steps; (1) the subject mean was first subtracted to center the data [34]; (2) data were normalized to the mean Euclidean distance [38,43]; and (3) the data were weighted according to the relative weight contribution represented by each marker [44]. The data sets from all participants were concatenated to create one 208,000 × 90 input matrix (lines: 26 participants, two trials of 16 s recorded at 250 Hz; columns: three-dimensional (3D) coordinates of 30 markers). The rows of the matrix contained the available posture information about the participant’s position at a certain time point and were interpreted as 90-dimensional posture vectors [33,45,46,47]. After performing the normalization and concatenation of the data from all volunteers and all trials, one PCA could be conducted for the whole dataset, allowing a direct comparison of the resultant movement components between all included trials and all participants. The PCA was calculated as eigenvector decomposition of the covariance matrix of the data. The resultant eigenvectors *PC_k_* (where *k* specifies the order of the eigenvector) thus form a new basis in the vector space of the posture vectors. As kinematic movement data were analyzed, each *PC_k_* describes one linear pattern of correlated marker movements. Two other output variables of the PCA are eigenvalues *EV_k_* and scores *PP_k_*(*t*). The scores represent the posture vectors in the new *PC_k_*-basis and are obtained through a basis transformation of the original data onto the new basis. Therefore, the scores can be interpreted as “principal positions” *PP_k_*(*t*), since they are a representation of positions in posture space [36,38,42,48,49]. The principal position *PP_k_*(*t*) together with the eigenvector *PC_k_* define one (the *k*-th) component of the whole postural movement. We call these movement components “principal movements” (*PM_k_*) as they describe linear representations of the involved movement dimensions. How much contribution each *PM_k_* provided to the overall postural variance across all participants is indicated by the *EV_k_* [33,45,46,47,49]. 

To perform further analyses with an analogue and subject-specific variable, we calculated the relative variances *rVAR_k_* from *PP_k_*(*t*). The *rVAR_k_* quantify how much each *PM_k_* contributed to the whole postural variance for each trial [34,48]. The variable *rVAR_k_* was evaluated to identify differences in movement strategy and movement dimensions between the trials and participants (Hypothesis 1). 

The idea of the *PP_k_* representing the positions of the body can be expanded by (double) differentiation to calculate the “principal accelerations” *PA_k_*(*t*) of each *PM_k_*. By counting the “number of zero crossings” (*N_k_*) in the *PA_k_* time-series, one obtains a variable to investigate the temporal characteristics of how the movement components are controlled (Hypothesis 2) [38]. Each *PA_k_* can be interpreted as the result of the interplay between agonistic and antagonistic muscle action. If more *N_k_* are present, a tighter movement control can be suggested as the sensorimotor system makes more frequent corrections. Reductions in *N_k_* may be related to longer processing times for postural adjustments or could indicate that a movement component is not controlled as tightly, for example, because the control of *PM_k_* is deprioritized [36,37,38]. 

### 2.5. Statistics

The two dependent variables in the current study were *rVAR_k_* to detect differences in movement strategy and *N_k_* to investigate alterations in the ‘tightness’ of motor control. Shapiro–Wilk tests were used to check for normal distribution of the data for each group, task, and movement component. 

The primary goal of the statistical analysis was to investigate effects of the between-subject factor “group” (adolescent/adult) and the within-subject factor “eye condition” (eyes closed/eyes open) and potential interaction effects on *rVAR_k_* and *N_k_*. Since *rVAR_k_* were not normally distributed, separate Mann–Whitney U tests were performed to compare *rVAR_k_* in each movement component *k* between adolescents and adults and between eye conditions. “Group”, “eye condition” and interaction effects of the normally distributed *N_k_* were tested using a repeated-measures MANOVA to implement all variables in one statistical model. In the case of significant main effects, subsequent t-tests were computed. The effect sizes were calculated using Rosenthals *r* for non-parametric tests [50] and partial eta square *η*_p_^2^ for parametric tests.

Since prerequisites for an ANCOVA calculation were not fulfilled, the secondary goal of determining potential effects of body height was investigated by splitting the adult group into two subgroups: (1) Adults smaller or equal than 180 cm (Adults ≤ 180) and (2) adults taller or equal than 190 cm (Adults ≥ 190). Effects of the new three-leveled “group” variable and of the within-subject factor “eye condition” on *rVAR_k_* and *N_k_* were analyzed similarly to the first approach: since *rVAR_k_* were not normally distributed, separate Kruskal–Wallis tests were performed. Effects on the normally distributed *N_k_* were tested using a repeated-measures MANOVA. 

The *p*-values corresponding to all post-hoc or pairwise comparisons for individual movement components were adjusted according to the Holm–Bonferroni correction to lower the risk of type-I error. The newly calculated thresholds were *p* < 0.0055, *p* < 0.0063, *p* < 0.0071, and *p* < 0.0083 in the first, second, third, and fourth rank, respectively. All statistical tests were performed using SPSS (IBM SPSS Statistics, Version 24, SPSS Inc., Chicago, IL, USA).

## 3. Results

### 3.1. PCA Results

The first nine analyzed movement components explained 98.37% of the overall movement variance during the balancing trials. A video sequence, which can be found in the Appendix A, contains visualizations of these movement components. The extreme positions of each component along with the respective eigenvalues *EV_k_* and qualitative descriptions are presented in in Figure 1.

### 3.2. Main Results—Motor Control Differences between Adolescents vs. Adults

#### 3.2.1. Coordinative Structure—rVAR_k_

In the eyes-open trial (Table 2), results were inconclusive. In three movement components (*PM*_4_, *PM*_5_, *PM*_9_), medium effect sizes and small *p*-values were observed, however, these *p*-values did not meet the threshold of significance after the Holm–Bonferroni correction. The results for the eyes-closed trials (Table 3) did not suggest a difference in the coordinative structure between adolescents and adults. The comparison of *rVAR_k_* between the eyes open and eyes closed condition (without Table) revealed *p*-values smaller than 0.05 in *rVAR*_2_ (*z* = −2.222, *p* = 0.026, *n* = 26) and in *rVAR*_6_ (*z* = −2.197, *p* = 0.028, *n* = 26), however, significance was not reached when applying the Holm–Bonferroni correction. 

#### 3.2.2. Temporal Control Characteristics—N_k_

When analyzing the number of zero crossings (*N_k_*), the group effect between adults and adolescents was significant (F(9,16) = 5.587, *p* = 0.001, *η*_p_^2^ = 0.759). Significant differences in *N_k_* were found in some but not in all *PM_k_* as listed in Table 4. Adults always presented a higher *N_k_* than adolescents except for *N*_1_. There was no significant group × eye condition interaction effect (F(9,16) = 0.381, *p* = 0.927, *η*_p_^2^ = 0.177), nor an eye condition effect (F(9,16) = 1.710, *p* = 0.167, *η*_p_^2^ = 0.490). 

### 3.3. Secondary Results—Differences between Adolescents, Smaller and Taller Adults

#### 3.3.1. Relative Variance—rVAR_k_

Preliminary differences between adolescents and small and tall adults (according to *p* < 0.05) could be observed in *rVAR*_3_, *rVAR*_5_ and *rVAR*_9_ but did not meet the threshold of significance after the Holm–Bonferroni correction. Except for the lower-order components, *rVAR*_1_ and *rVAR*_2_, adolescents appeared on average to contribute more *rVAR_k_* compared to smaller adults while there was little difference compared to taller adults (Table 5). In the eyes-closed trial (without Table), no significant differences were found.

#### 3.3.2. Number of Zero-Crossings—N_k_

The group effect between small adults, tall adults and adolescents was statistically significant (F(18,32) = 2.943, *p* = 0.004, *η*_p_^2^ = 0.623). Individual comparisons are presented in Table 6 and visualized in Figure 2. There was no significant group × eye condition interaction effect (F(18,32) = 0.892, *p* = 0.591, *η*_p_^2^ = 0.334), nor an eye condition effect (F(9,15) = 1.823, *p* = 0.146, *η*_p_^2^ = 0.522).

## 4. Discussion

### 4.1. Main Results—Motor Control Differences between Adolescents vs. Adults

The current study investigated alterations in the coordinative structure of postural movements and in the control of individual movement strategies between adolescence compared to adulthood. Considering our first hypothesis, the expectation of differences in movement strategies between adolescents and adults, which would mostly manifest in anterior–posterior direction, can only be partially supported. Some significant differences (*p* < 0.05) were identified between conditions for a number of relative variance variables. However, after controlling for alpha error, the statistical significance disappeared. However, results are in accordance with our second hypothesis; the expectation that less tight control would be shown by adolescents. This was supported by the differences in the *N_k_* between adolescents and adults for movement components *PM*_2_, *PM*_5_ and *PM*_6_. It should be noted that these *PM_k_* describe movement components predominantly representing motion in anterior–posterior direction, as it was hypothesized. This finding is strengthened by the fact that it remained significant even with a relatively conservative correction for multiple comparisons. 

Showing differences in selected *PM_k_* but not in all analyzed *PM_k_* agrees well with the minimum intervention principle [23,24]. It is stated that motor control intervenes if task-relevance is at risk but allowing movement variability as long as the primary goal is not affected negatively. Considering that the significances appeared in *PM_k_* representing movements in anterior–posterior direction, which is suggested to be more stable in tandem stance than the medio-lateral direction and therefore is less task-relevant to control, results appear to be conclusive. Moreover, they agree with previous studies on effects in postural control, again revealing differences in some but not all *PM_k_*, e.g., due to aging [38]. Contrary to results in the current study, older individuals presented differences especially in task-relevant movement components like medio-lateral ankle sway [38], which can be interpreted as a sign of changes in sensorimotor control due to e.g., degeneration of structures [51]. On the other hand, the differences in our results might be driven by the still developing automatisms in adolescents [13]. Adolescents may control task-relevant movement components in a rather reliable way; however, redundant components show alterations leading to the observed variability. Results are further in line with previous studies investigating leg dominance [36,37,52], sensory perturbation [48], or dual tasking [39] and its effect on postural control, presenting some but not all *PM_k_* to be affected. Although adolescents are suggested to heavily rely on visual information [13], no differences could be observed between the eyes-open and eyes-closed trial. Since the trials were not randomized, a learning or training effect leading to habituation effects might have occurred which would be explainable by the concept of exploration (a concept well established in perception-action theory [53,54]). Then, however, the absence of any “eye condition” effect could be interpreted as a result of exploration, which in this case seems to be sufficient to maintain balance. Since our study design is not well-suited to answer this question, this assumption remains speculative.

The altered activity in anterior–posterior movement components in adolescents’ tandem stance confirms the expectation that exploration would be increased in adolescents. To the knowledge of the authors, this study was the first to hypothesize that exploration patterns would follow a pattern predictable by optimal feedback control theory. This raises the question whether these theories are of compatible nature. It is a general limitation of optimal control theory that control laws are only defined during movement; perhaps perception-action theory could be used as a supplemental theory to answer the question of how we decide to engage in these movements [53,55,56]. In contrast, the control laws defined by optimal control theory can provide insight into the weighting of alternative movement plans for ongoing action, a process similar to what has been hypothesized in the affordance competition hypothesis [57]. More work is required to determine the feasibility of combining these conceptual frameworks, which could be in the field of both hypothesis testing and more philosophical work. 

### 4.2. Secondary Results—Differences between Adolescents, Smaller and Taller Adults

It was a secondary purpose of this study to analyze whether effects that had been attributed to differences between adolescence and adulthood could also be explained by body height. Thereto, the adult groups were split into two subgroups based on height. Independently of the sensory situation (eyes open or eyes closed), we found that all three significant effects that we had observed between adolescents and the group of all adults (in *N*_2_, *N*_5_ and *N*_6_) could be explained by differences between adolescents and the smaller adults. In fact, we found an additional effect between groups in *N*_7_. Further, when comparing the number of zero crossings between the three groups (Figure 2), we found that in all movement components, with the exception of *PM*_1_, that the *N_k_* of both the adolescents and the tall adults were smaller than those of the smaller adults. This suggests that body height might play a role in the control of *PM*_1_, but developmental stage is more important for the observed differences in postural control in the other *PM_k_*. We would reason that very tall individuals sometimes also demonstrate forms of ‘motor awkwardness’; in this sense, it is fitting that we find a similar pattern in tall adults and adolescents in our data, although this suggestion is speculative since the differences between tall and less-tall adults were not significant in the current study. 

In *PM*_1_, which is the largest movement component representing 60% of the postural variance, both variables relating to relative variance (*rVAR*_1_) and tightness of control (*N*_1_) showed changes that would be consistent with the differences in body height (an increase in *rVAR*_1_ and a decrease in *N*_1_ with increasing height), but these differences were not significant. Thus, if body height is assumed to affect postural control during tandem stance, then *rVAR*_1_ and *N*_1_ would be the suitable variables for further investigations. 

Finally, we assessed differences in postural control between adolescents and adults when standing with eyes open vs. eyes closed. Previous research had suggested that adolescents rely more on visual sensory information due to a lack of automatized control strategies [13], therefore, different results were expected between the eyes-open and eyes-closed trials. However, the prediction was not confirmed by our results. 

### 4.3. Limitations

The relatively small sample size and subsequent division into three groups is a limitation of the current study. Still, significant differences could be observed and effect sizes varied from medium to very large (Table 2, Table 3, Table 4, Table 5 and Table 6) in several of the movement components. This limitation affects particularly the analysis related to body height, as we only had five adults in the ‘tall group’. However, the observation that *p*-values did not increase despite the now smaller sample size is, in our opinion, good evidence for the assertion that the age differences are not driven by differences in body height. Nevertheless, further research with larger sample size is required to examine whether there are effects of body height. 

A general limitation to the current study comes from unclarity over definitions of ‘awkwardness’ available within the research field. We cannot be sure that all adolescent participants were influenced by the phenomenon of adolescent awkwardness, since there is no definition or test for adolescent awkwardness. A longitudinal design would be recommended for a future follow-up study to monitor the maturation process and the associated development of postural control characteristics. 

We did not randomize the order of the eyes-open and eyes-closed trials to avoid falls of the participants and the data recording varied in time for the trials. As a consequence, habituation and motor acquisition effects cannot be ruled out, which is a limitation when comparing these conditions.

With the methodological approach used in the current study, over 98% of the postural variance was captured within the nine analyzed principal movements. This indicates the representativeness of the approach to the actual behavior. It could be reasoned that more principal movements would result in an even better fit, however, the authors decided against analyzing more components as this would reduce the interpretability of the results (and all further components explain less than 0.25% of the variance per individual component). However, this implies that the results are an approximation of the actual postural movements, as the whole set of *PM_k_* would be necessary to fully reconstruct the original motion.

## 5. Conclusions

The current study could find differences in the temporal structure of how postural movement components are controlled between adolescents and adults. These differences were in line with the hypothesis of slower processing of motor commands in adolescents as a phenomenon of “adolescent awkwardness”. Our results further suggest that the observed differences in motor control are not a result of the smaller body height of adolescents; on the contrary, tall adults showed in some movement components similar control characteristics to the adolescent group. 

## Figures and Tables

**Figure 1 brainsci-10-00216-f001:**
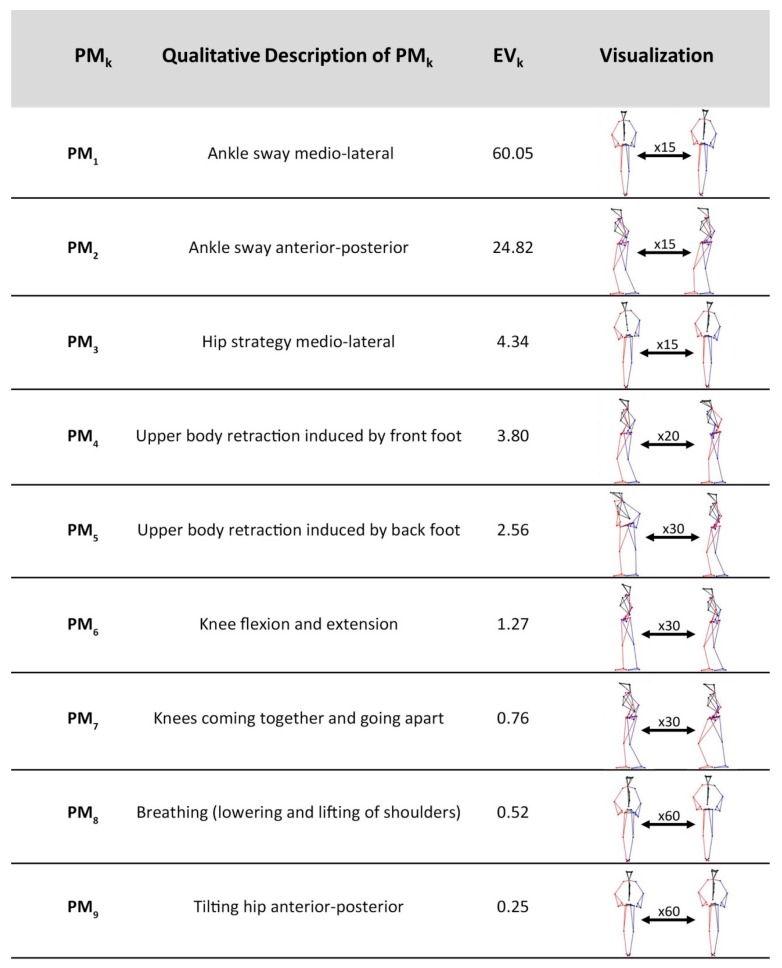
Qualitative description of the first nine movement components (*PM_k_*). The *Eigenvalues EV_k_* indicate the contribution of each *PM_k_* to the overall movement; the visualization of the components was amplified by factor *x*.

**Figure 2 brainsci-10-00216-f002:**
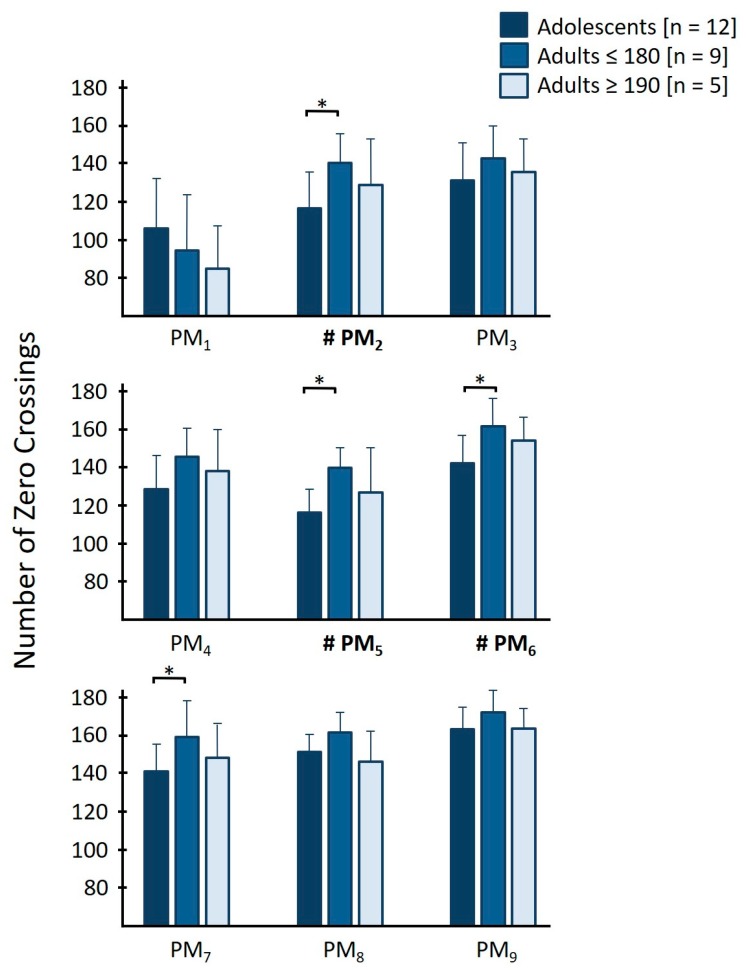
Visualization of the differences in “number of zero crossings” between adolescents (dark blue), adults smaller or equal than 180 cm (blue) and adults taller or equal than 190 cm (light blue) in the different *PM_k_*. Bold *PM_k_* with hashtag (#) indicate statistically significant main effects, asterisks (*) between the bars indicate significant post-hoc differences between groups.

**Table 1 brainsci-10-00216-t001:** Anthropometric differences between the participants.

Participants	Adolescents *n* = 12	Adults ≤ 180 cm *n* = 9	Adults ≥ 190 cm *n* = 5	*p*-Value ANOVA
**Age [years]**	13.2 ± 0.9	25.4 ± 3.1	27.4 ± 2.2	<0.001
**Height [cm]**	168.1 ± 8.8	177.9 ± 3.0	192.0 ± 2.5	<0.001
**Weight [kg]**	59.8 ± 10.4	74.2 ± 4.1	87.8 ± 8.0	<0.001

**Table 2 brainsci-10-00216-t002:** *rVAR_k_* of adolescents and adults in the different movement components during the eyes-open trial. Means ± SD of each group are presented, bold *p*-values present *p*-values < 0.05, which did not meet the threshold of significance after correction. Effect sizes (*r*) are displayed according to Rosenthal.

*rVAR_k_*	Adolescents Mean	Adults Mean	*p*-Value	*r*	Cumulative Adolescents	Cumulative Adults
***rVAR*_1_**	51.2 ± 22.6	62.2 ± 14.5	0.274	0.222	51.2	62.2
***rVAR*_2_**	31.1 ± 19.9	28.3 ± 14.9	0.820	0.050	82.3	90.5
***rVAR*_3_**	4.9 ± 3.9	2.8 ± 2.3	0.131	0.303	87.2	93.3
***rVAR*_4_**	5.2 ± 6.0	2.2 ± 1.6	**0.036**	0.414	92.4	95.5
***rVAR*_5_**	2.8 ± 1.8	1.4 ± 1.5	**0.011**	0.494	95.2	96.9
***rVAR*_6_**	1.4 ± 1.5	0.8 ± 0.6	0.322	0.202	96.6	97.7
***rVAR*_7_**	0.7 ± 0.4	0.6 ± 0.5	0.274	0.222	97.3	98.3
***rVAR*_8_**	0.7 ± 1.0	0.3 ± 0.2	0.145	0.293	98.0	98.6
***rVAR*_9_**	0.3 ± 0.2	0.2 ± 0.3	**0.041**	0.403	98.3	98.8

**Table 3 brainsci-10-00216-t003:** *rVAR_k_* of adolescents and adults in the different movement components during the eyes-closed trial. Means ± SD of each group and corresponding *p*-values are presented. Effect sizes (*r*) are displayed according to Rosenthal.

*rVAR_k_*	Adolescents Mean	Adults Mean	*p*-Value	*r*	Cumulative Adolescents	Cumulative Adults
***rVAR*_1_**	61.0 ± 19.3	61.9 ± 18.4	0.940	0.020	61.0	61.9
***rVAR*_2_**	21.5 ± 19.4	23.5 ± 15.0	0.252	0.232	82.5	85.4
***rVAR*_3_**	5.0 ± 3.7	4.0 ± 5.3	0.231	0.242	87.5	89.4
***rVAR*_4_**	4.4 ± 4.8	3.4 ± 4.7	0.462	0.151	91.9	92.8
***rVAR*_5_**	3.4 ± 2.5	2.5 ± 2.3	0.193	0.262	95.3	95.3
***rVAR*_6_**	1.7 ± 1.6	1.5 ± 1.7	0.705	0.081	97.0	96.8
***rVAR*_7_**	0.7 ± 0.4	1.0 ± 1.0	0.494	0.141	97.7	97.8
***rVAR*_8_**	0.5 ± 0.3	0.5 ± 0.3	0.595	0.111	98.2	98.3
***rVAR*_9_**	0.3 ± 0.2	0.2 ± 0.2	0.432	0.161	98.5	98.5

**Table 4 brainsci-10-00216-t004:** *N_k_* of adolescents and adults in the different movement components. Results from eyes-open and eyes-closed conditions were pooled due to the absent eye condition effect. Means ± SD of each group and corresponding *p*-values are presented. Bold values indicate *p*-values < 0.05; asterisks (*) mark significant differences after Holm–Bonferroni correction. Effect sizes are displayed as *η*_p_^2^ (partial eta-square).

*N_k_*	Adolescents Mean	Adults Mean	*p*-Value Group	*η* _p_ ^2^
***N*_1_**	105.9 ± 24.3	89.0 ± 27.0	0.119	0.098
***N*_2_**	116.0 ± 19.8	134.4 ± 19.7	**0.004 ***	0.298
***N*_3_**	131.7 ± 18.8	139.0 ± 18.0	0.200	0.068
***N*_4_**	129.7 ± 17.4	143.0 ± 17.2	**0.020**	0.206
***N*_5_**	117.8 ± 10.8	134.2 ± 16.5	**0.001 ***	0.350
***N*_6_**	143.9 ± 13.1	159.2 ± 13.6	**0.001 ***	0.379
***N*_7_**	142.3 ± 14.1	154.7 ± 17.3	**0.026**	0.191
***N*_8_**	152.5 ± 9.6	154.6 ± 15.1	0.355	0.036
***N*_9_**	164.3 ± 10.7	169.1 ± 11.1	0.122	0.097

**Table 5 brainsci-10-00216-t005:** Group differences of *rVAR_k_* in the different movement components during the eyes-open trial are presented. On the left, means ± SD of the adolescents and smaller and taller adults are shown. On the right, *p*-values resulting from the Kruskal–Wallis tests can be seen, bold *p*-values indicate *p* < 0.05, which, however, did not meet the threshold of significance after correction. Effect sizes are displayed as *η*_p_^2^ (partial eta-square).

*rVAR_k_*	Adolescents Mean	Adults ≤ 180 Mean	Adults ≥ 190 Mean	*p*-Value	*η* _p_ ^2^
***rVAR*_1_**	51.2 ± 22.6	58.8 ± 14.1	68.2 ± 14.7	0.345	0.445
***rVAR*_2_**	31.1 ± 19.9	34.1 ± 13.6	17.9 ± 9.7	0.125	0.094
***rVAR*_3_**	4.9 ± 3.9	1.6 ± 1.3	5.0 ± 2.2	**0.011**	0.303
***rVAR*_4_**	5.2 ± 6.0	1.7 ± 0.9	3.0 ± 2.4	0.054	0.167
***rVAR*_5_**	2.8 ± 1.8	0.8 ± 0.1	2.4 ± 2.3	**0.021**	0.248
***rVAR*_6_**	1.4 ± 1.5	0.8 ± 0.1	0.8 ± 0.5	0.569	0.038
***rVAR*_7_**	0.7 ± 0.4	0.4 ± 0.1	0.9 ± 0.8	0.124	0.095
***rVAR*_8_**	0.7 ± 1.0	0.3 ± 0.1	0.3 ± 0.1	0.254	0.032
***rVAR*_9_**	0.3 ± 0.2	0.2 ± 0.1	0.2 ± 0.1	**0.038**	0.197

**Table 6 brainsci-10-00216-t006:** Group differences of *N_k_* in the different movement components are presented. Results from eyes-open and eyes-closed conditions were pooled due to the absent eye condition effect. The means ± SD of the adolescents, small (≤180 cm) and tall (≥190 cm) adults and corresponding *p*-values are displayed. Bold *p*-values indicate *p* < 0.05, asterisks (*) mark significant differences after Holm-Bonferroni correction. Effect sizes are presented as *η*_p_^2^ (partial eta-square). *p*-values related to post-hoc test between all three groups are displayed on the right.

*N_k_*	Adolescents Mean	Adults ≤ 180 Mean	Adults ≥ 190 Mean	*p*-Value	*η* _p_ ^2^	Adolescents Adults ≤ 180	Adolescents Adults ≥ 190	Adults ≤ 180 Adults ≥ 190
***N*_1_**	105.9 ± 24.3	94.2 ± 29.2	83.7 ± 22.2	0.227	0.121	0.873	0.306	1.000
***N*_2_**	116.0 ± 19.8	140.2 ± 17.0	128.5 ± 22.7	**0.007 ***	0.348	0.006 *	0.448	0.580
***N*_3_**	131.7 ± 18.8	142.2 ± 19.2	135.9 ± 15.8	0.351	0.087	0.458	1.000	1.000
***N*_4_**	129.7 ± 17.4	146.5 ± 14.6	139.5 ± 21.2	**0.012**	0.322	0.011	1.000	0.179
***N*_5_**	117.8 ± 10.8	140.8 ± 10.1	127.7 ± 22.4	**0.002 ***	0.429	0.001 *	0.385	0.265
***N*_6_**	143.9 ± 13.1	163.4 ± 13.5	155.0 ± 12.6	**0.001 ***	0.475	0.001 *	0.352	0.154
***N*_7_**	142.3 ± 14.1	159.9 ± 17.1	149.4 ± 16.4	**0.001 ***	0.440	0.002 *	1.000	0.012
***N*_8_**	152.5 ± 9.6	162.5 ± 10.6	146.6 ± 17.2	**0.024**	0.277	0.114	0.878	0.033
***N*_9_**	164.3 ± 10.7	173.3 ± 11.2	164.9 ± 9.3	0.086	0.192	0.109	1.000	0.339

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
