# Peer review of "Adolescent Awkwardness: Alterations in Temporal Control Characteristics of Posture with Maturation and the Relation to Movement Exploration"

_brainsci, 2020, doi:10.3390/brainsci10040216_

Round 1
Reviewer 1 Report
Summary
This study tested two hypotheses; adolescents control their movements less tightly, and use a coordinative structure comparing different from the adult. Moreover, they tested whether the difference between the adolescents and adults is due to body height alterations. PCA was used for the data analysis. rVAR estimated the difference in the coordinative structure between the adolescents and adults. rVAR was not significantly different between the groups and between the visual input conditions, indicating that their hypothesis was not supported. Tightness of motor control (temporal control characteristics) was estimated by N. The results of the N revealed that some Ns were significantly different between the groups, partially supporting their hypothesis that a tendency that the adolescents control movement less tightly. Body height was not cause of the difference in the motor control between the groups, indicating that the finding did not support their hypothesis. Basically, this manuscript is well written. In spite of that, I found some concerns as showing below.
Major concern 1.
One major concern is that small number of the tall adults (>190 cm), as the authors mentioned in Limitation. This enlarges the risk of the type II error of the statistical analysis. I recommend the authors to increase the sample size of the adult group or to eliminate the results in secondary goal (3.3 Secondary goal) that divided the adult group into two.
Major concern 2.
The trial, in which the eyes were closed, was always conducted after the first trial in which the eyes were opened. Through experiencing trials, habituation or motor acquisition may have occurred. Accordingly, it is not possible to rule out the possibility that the first and second trials are under unequal conditions because of those potential independent variables. Thus, the visual input factor (eye-closed vs. eyes-opened) is not comparable in this study. The two conditions must have been counterbalanced across the participants.
Major concern 3.
The time taken for the trial with eyes opened and that for the trial with eyes closed were not measured under equal time; the time was taken 60 s in the first trial, but that was taken 30 s in the second trial. We do not know what happens in the time window 31-60 s after the start of the trial. Thus, this study design is also the reaspon that the first trial with eyes opened and the second trial with eyes closed are not comparative.
Minor concerns
L102-103
The PCA has been spelled out in L92.
L208
I do not understand why the authors did not use ANOVA for the statistical analyssi. Please explain.
L222-223
Adult group was divided into the participants taller than 190 cm and those shorter than 180 cm. How about the participants between 180 – 190 cm ?
L306
In Discussion, only two subheadings were given. I recommend the authors to divide “Main Results” section into two or three sections for improving readability.
Table 6
In the caption for this table, they indicate “bold p-value”, but no bold p-values appear in this table.
Reviewer 2 Report
Abstract
- No comments
Introduction
- Lines 61-62: Variability in postural stance is also useful for having the ability to adapt current actions to cope with unexpected changes in the environment (see literature on flexibility in motor actions: e.g., Loverro, K. L.*, Laudicina, N. M., Gill, S. V. & Lewis, C. L. (2016). Changes in gait with anteriorly added mass: A pregnancy simulation study. Journal of Applied Biomechanics, 32: 378-87).
- Lines 73-75: I’m assuming that the tandem stance example in which increased exploration would occur in the anterior/posterior direction only holds true if adolescents are not allowed to exhibit compensatory strategies (e.g., hold their arms out to their sides to increase medial/lateral stability). Is this correct?
- Line 95: Are “interventions” strategies?
Method
- Lines 121-6 should be placed in procedure because I presume that these were instructions that participants received. If not, then they should be included in data processing.
Results
- No comments
Discussion
- No comments
Round 2
Reviewer 1 Report
Thank you for the revision.
There is no further comment on this version of the manuscript.